# pH-Dependent Structural Dynamics of Cathepsin D-Family Aspartic Peptidase of *Clonorchis sinensis*

**DOI:** 10.3390/pathogens10091128

**Published:** 2021-09-02

**Authors:** Jung-Mi Kang, Hương Giang Lê, Byoung-Kuk Na, Won Gi Yoo

**Affiliations:** 1Department of Parasitology and Tropical Medicine, Institute of Health Sciences, Gyeongsang National University College of Medicine, Jinju 52727, Korea; jmkang@gnu.ac.kr (J.-M.K.); gianglee291994@gmail.com (H.G.L.); bkna@gnu.ac.kr (B.-K.N.); 2Department of Convergence Medical Science, Gyeongsang National University, Jinju 52727, Korea

**Keywords:** *Clonorchis sinensis*, cathepsin D, aspartic peptidase, molecular dynamics simulation, pH effect, flap dynamics

## Abstract

Cathepsin D (CatD; EC 3.4.23.5) family peptidases of parasitic organisms are regarded as potential drug targets as they play critical roles in the physiology and pathobiology of parasites. Previously, we characterized the biochemical features of cathepsin D isozyme 2 (CatD2) in the carcinogenic liver fluke *Clonorchis sinensis* (CsCatD2). In this study, we performed all-atomic molecular dynamics simulations by applying different systems for the ligand-free/bound forms under neutral and acidic conditions to investigate the pH-dependent structural alterations and associated functional changes in CsCatD2. CsCatD2 showed several distinctive characteristics as follows: (1) acidic pH caused major conformational transitions from open to closed state in this enzyme; (2) during 30–36-ns simulations, acidic pH contributed significantly to the formation of rigid β-sheets around the catalytic residue Asp_219_, higher occupancy (0% to 99%) of hydrogen bond than that of Asp_33_, and enhanced stabilization of the CsCatD2-inhibtor complex; (3) neutral pH-induced displacement of the N-terminal part to hinder the accessibility of the active site and open allosteric site of this enzyme; and (4) the flap dynamics metrics, including distance (*d*_1_), TriCα angles (*θ*_1_ and *θ*_2_), and dihedral angle (ϕ), account for the asymmetrical twisting motion of the active site of this enzyme. These findings provide an in-depth understanding of the pH-dependent structural dynamics of free and bound forms of CsCatD2 and basic information for the rational design of an inhibitor as a drug targeting parasitic CatD.

## 1. Introduction

Clonorchiasis is a parasitic disease caused by the liver fluke *Clonorchis sinensis*. This parasite is prevalent in far East Asian countries, including China, Korea, and Northern Vietnam, and it is reported to infect approximately 35 million people worldwide [1]. Infections in humans usually occur because of consumption of raw or inadequately cooked freshwater fish carrying the *C. sinensis* metacercariae. After the infection, the metacercariae exist in the duodenum of the human body and the juvenile worms migrate through the ampulla of Vater to reach the common bile duct. *C. sinensis* worms induce a series of pathological changes in the bile duct, resulting in epithelial hyperplasia, periductal fibrosis, obstructive jaundice, dyspepsia, and cirrhosis of the liver [2]. Chronic clonorchiasis can induce diverse complications, such as periductal inflammation, fibrosis, cholangitis, cholelithiasis, and cholangiectasis [1,3,4]. *C. sinensis* has been classified by the World Health Organization as Group I biocarcinogen that promotes cholangiocarcinoma in humans [5].

Peptidases of parasitic helminths are one of the most extensively studied molecules because of their pivotal pathobiological roles in the physiology and nutrition of parasites as well as host–parasite interactions [6,7,8,9,10,11,12,13,14,15]. Among various peptidases, cathepsin D (CatD; EC 3.4.23.5) is considered an attractive target for the design of vaccines or anthelminthic drugs [16,17,18,19], as it is an essential component for the degradation of the host hemoglobin or proteins within the gut of blood-feeding flukes [18,20,21,22]. Recently, we characterized the biochemical and immunological properties of two CatDs of *C. sinensis*, cathepsin D isozyme 1 (CsCatD1) and cathepsin D isozyme 2 (CsCatD2) [23]. CsCatD2 is likely to play an important role as a digestive enzyme as it is mainly expressed in the intestinal epithelial cells of *C. sinensis* adult worms. In light of its importance in parasite survival and nutrition, inhibition of the CsCatD2 may deliver an anti-clonorchiasis effect although the physiological role of CsCatD in *C. sinensis* has yet to be elucidated.

Enzyme–substrate binding is a dynamic process in which enzymes undergo structural changes, resulting in the disruption of chemical bonds in the substrates and/or formation of new ones to yield the end products [24]. The active site pocket of typical CatDs is covered by the Y flap region (commonly known as the antiparallel β-sheet) that plays an important role in binding of the substrate [25]. Understanding the flap dynamics of CatDs is essential to gain an in-depth knowledge of the molecular aspects of enzyme-substrate interactions, particularly in pH-dependent structural alterations of the enzyme. It may also provide crucial information for the design of optimized aspartic peptidase inhibitors as drugs targeting the CatDs of the parasite.

In this study, we performed all-atomic molecular dynamics (MD) simulations in the ligand-free and bound forms of the CsCatD2 homology models under acidic and neutral conditions to explore the molecular features of flap dynamics and study the pH-induced structural alterations of CsCatD2.

## 2. Results and Discussion

### 2.1. Homology Models and Features of Mature CsCatD2

The models for free-CsCatD2 (CsCatD2_free_) and bound-CsCatD2 (CsCatD2_bound_) were built based on the experimental X-ray characterized structures of *Ixodes ricinus* cathepsin D 1 (IrCD1), without the inhibitor (Protein Data Bank (PDB) code: 5n7n) and in complex with the pepstatin (PDB code: 5n7q) from the tick *I. ricinus* (Appendix A). The IrCD1 structures were chosen as the best templates with an E-value of 4e-148, sequence similarity of 73% and coverage of 96% via the protein-Basic Local Alignment Search Tool (BLASTP) against the PDB database [26]. To generate the CsCatD2_bound_ structure model, the pepstatin from the template (PDB code: 5n7q) was superimposed on the homology model of CsCatD2 during the modeling process. Mature CsCatD2 showed a conserved bilobed structure consisting of two antiparallel β-sheet domains packed against each other. The interdomain connects the N- and C-terminal domains and each contributes one catalytic aspartate residue, aspartate 33 (Asp_33_) and aspartate 219 (Asp_219_), respectively, thereby forming the active site pocket. The Y flap region and polyproline loop covered this active site pocket to facilitate substrate binding (Figure 1A). Glycine 79 (Gly_79_) was situated as a flap tip residue, whereas methionine 297 (Met_297_) was placed as a hinge residue on the opposite side.

Pepstatin is an inhibitor of both subfamilies of aspartic proteases, A1 and A2 [27]. To explore potential anti-*C. sinensis* compounds targeting CsCatD2, it is necessary to understand the structure of mature CsCatD2 complexed with pepstatin. Binding of the inhibitor to the active site of CsCatD2 induced conformational changes in the flap region containing conserved Gly_79_ and the polyproline loop, which recoiled toward the inhibitor. Between CsCatD2_free_ and CsCatD2_bound_, the tip-to-tip distance and the hinge-to-hinge distance at pH 4.0 were 4.98 Å and 1.5 Å, respectively (Figure 1B).

### 2.2. Four Systems for MD Simulations According to the Enzyme Conformations and pH Values

The proteolytic activity of CatDs is generally optimum at low pH and can be detected at neutral pH for a limited period of time [28]. There have been several reports regarding the optimal pH for a CatD enzymatic activity which are mentioned as follows: *Trichomonas vaginalis* CatD proteolytic activity is high at an acidic pH (3.5–4.5) and low at a pH close to 7 [29]; optimal pH of HsCatD is in the pH range 3.0–4.5 [30]. Considering the optimal pH ranges studied previously, we selected acidic (pH 4) and neutral (pH 7) conditions to assess the influence of different pH values on the stability and flexibility of CsCatD2. In addition, it was reported that a 50 nanosecond (ns)-MD simulation is a sufficiently long simulation time to observe all the conformational changes of aspartic peptidases [31,32]. The MD trajectories were analyzed in terms of stability, flexibility, and compactness, including the root mean square deviation (RMSD), root mean square fluctuation (RMSF), radius of gyration (Rg), and solvent-accessible surface area (SASA).

### 2.3. CsCatD2_free_ Adopts a Semi-Closed Conformation for Ligand-Binding at Acidic pH

The conformational stability of CsCatD2_free_ was evaluated by calculating the backbone RMSD by superposing the MD trajectories onto each initial structure at 0 ns. Simulations at the bound states were stable regardless of pH, as evidenced by low and relatively constant RMSD values (1.48 ± 0.01, pH 4, and 1.53 ± 0.01, pH 7) of the trajectories (Figure 2A). CsCatD2_free_ was relatively unstable at pH 7 (average RMSD: 2.36 ± 0.02) throughout the simulation, which was revealed by the deflection at the beginning of the simulation. However, it remained stable at pH 4 (average RMSD: 1.90 ± 0.02), comparable to the RMSD level of the bound form until approximately 32.0–35.5 ns. At these specific points-of-time, the RMSD was abruptly increased by 0.8 Å during which a transition in the tertiary structure may have occurred.

To better understand pH-specific conformational change during 30–35 ns, 28 snapshots from each system at 30–36 ns were analyzed using principal component analysis (PCA). Overall, the PCA plot visually indicated that CsCatD2_free_ was separated from CsCatD2_bound_ along the principal component 1 (PC1), accounting for 88.05% of the ligand-induced conformational changes while PC2 explains 4.66% of the pH-dependent variations (Figure 3A and Appendix A). Two snapshots of pH 4 were close to the ones of pH 7 in the CsCatD2_bound_ and it can contribute to the low score of PC2. However, all snapshots of pH 4 were distinctly far away from the ones of pH 7 for CsCatD2_free_ and, hence, PC2 can explain well the large variations of CsCatD2_free_ from 30 ns to 36 ns. The individual residue contribution to PC2 was examined in more detail by visual inspection (Figure 3B). Clamp hairpin (aa 158–168) between β-strand 7 (β_7_) and β_8_ showed the highest contribution to the PC2. N-terminal end, two hairpins including aa 284–287 between β_14_ and β_15_, and aa 266–269 region between β_12_ and β_13_ also contributed significantly to the PC2. Remarkably, N-terminal end tended to shift to open the active site and also the distance between the flap tip and hinge residue was gradually decreased in CsCatD2_free_ at pH 4 at different time-points 32.0, 33.50, and 34.25 ns (Appendix A). The RMSD was then maintained at a steady level upon adopting a new conformation, as supported by the increasing number of hydrogen bonds in CsCatD2_free_ at pH 4 (Figure 2A and Appendix A). Secondary structure elements showed considerable conformational change according to the pH effect and simulation time. In particular, β_11_ including Asp_219_, β_12_, and β_13_ formed only at pH 4. These results suggest that acidification of 30–36-ns simulation can contribute to major conformational changes in CsCatD2_free_ from open to closed/semi-closed conformations.

### 2.4. Both Ligand-Binding and Acidic pH Limit the Conformational Flexibility

RMSF provides insights regarding the flexible or disordered regions of molecules in a biological system during MD simulations [33]. In this study, residual atomic fluctuations in all systems showed similar trends below 4 Å throughout the course of the simulations, except for CsCatD2_free_ at pH 7 (Figure 2B). Overall, the free forms at pH 7 (average RMSF: 1.53) and pH 4 (average RMSF: 1.44) showed higher flexibility than bound forms at both pH values (average RMSF: 1.11). Two regions (aa 162–163 and aa 284–285) exhibited larger deviations of up to 7 Å from its starting structure in CsCatD2_free_ at pH 7 than at pH 4 (Figure 2C,D). _162_DV_163_ lies in the hairpin of the clamp region between the N- and C-termini, whereas _284_YR_285_ belongs to the β-loop region and is located near the polyproline loop. The results are in broad accordance with PCA analysis of 30–36-ns simulation, which exhibits high residual contributions of the clamp’s hairpin and β-loop region. This is also consistent with a previous report that flexibility occurs in some regions of free HsCatD owing to the absence of ligands [32]. Thus, it may be suggested that pH itself can affect major conformational fluctuations in CsCatD2_free_ as shown by the increase in pH toward neutral conditions, which caused great fluctuations in the two regions in CsCatD2_free_.

### 2.5. Neutral pH May Induce the Opening of Potential Allosteric Inhibitory Sites

Based on the average B-factor models for all systems, the N-terminal part covered the entrance of the active site in CsCatD2_free_ (Figure 4A,B), whereas there was no steric hindrance involving the entrance in the CsCatD2_bound_ (Figure 4C,D). In the free forms, three flexible regions (N-terminus, aa 162–163, and aa 284–285) were detected and the latter two regions showed higher mobility at pH 7 than at pH 4 in the bound forms as shown by the RMSF values (Figure 4A,B). For CsCatD2_free_, as described above, no hydrogen bond formation of the flap tip Gly_79_ (average RMSF: 1.55) with the inhibitor can have a critical impact on the overall flexibility and polyproline loop as compared to the CsCatD2_bound_ (average RMSF: 0.66, Gly_79_) under both pH conditions. These findings are consistent with the fact that the flexible loop folds inward over the active site and interacts with the flap as the HsCatD closes [32,34].

Remarkably, the region flanking aa 149–152 exhibited high flexibility only in CsCatD2_bound_ at pH 7 (Figure 4C). This highly mobile region was located at the boundary of the potential allosteric pocket, all of which were matched between CsCatD2 (glycine 14 (Gln_14_), tyrosine 15 (Tyr_15_), tyrosine 16 (Tyr_16_), phenylalanine 32 (Phe_32_), alanine 97 (Ala_97_), valine 149 (Val_149_), glutamine (Glu_170_), isoleucine 171 (Ile_171_), and phenylalanine 173 (Phe_173_)) and IrCatD1 (valine 39 (Val_39_), tyrosine 40 (Tyr_40_), tyrosine 41 (Tyr_41_), phenylalanine 57 (Phe_57_), alanine 120 (Ala_120_), leucine 172 (Leu_172_), glutamine 193 (Glu_193_), valine 194 (Val_194_), and phenylalanine 196 (Phe_196_)) with a template modeling (TM) score of 93% (Appendix A). These findings together with recent reports indicating that the allosteric mechanism regulates pepsin-family peptidases [28,35] suggest that neutral pH can induce the opening of the allosteric site of CsCatD2 similar to HsCatD and IrCatD [28,36]. Recently, propeptide-derived inhibitor allosterically inhibits IrCatD activity [28]. The allosteric mechanism for the peptidyl inhibitor is strikingly different as the potency of the allosteric inhibitor is pH-dependent. A pH shift toward neutral condition displaces the N-terminus of mature CatDs into the active site and opens the allosteric site. Thus, an inhibitor (pepstatin) may preferentially bind to the active site of CatDs at an acidic pH, whereas allosteric inhibitor (peptidyl inhibitor) favorably binds to an allosteric inhibitory site.

### 2.6. Acidic pH Enhances the Compactness of CsCatD2_free_ to Converge to the Level of Bound Forms

To monitor the compactness of CsCatD2_free_ and CsCatD2_bound_ structures, Rg and SASA were analyzed as the indicators of protein structure compactness in all systems. Rg indicates the mass-weighted RMSD between the common center-of-mass and a collection of atoms [37]. Although the low Rg values of CsCatD2_bound_ were consistent regardless of pH, the Rg values of CsCatD2_free_ tended to decline (Figure 5A). Remarkably, the Rg value of CsCatD2_free_ at pH 4 approached close to the level of CsCatD2_bound_ and eventually converged at approximately 44 ns. These results were further supported by the SASA value, which measures the exposed surface of the entire enzyme that is accessible to solvent molecules [38]. The overall trend of the SASA corresponded to fluctuations in the Rg values for all systems. At approximately 44 ns, the convergence of the SASA in CsCatD2_free_ at pH 4 occurred at the level of CsCatD2_bound_ (Figure 5B). These processes were in line with the abrupt changes in the RMSD values. The low Rg and SASA values throughout the remainder of the simulations were also consistent with the adoption of a new conformation.

### 2.7. Acidic pH Alters Correlated Motions and Hydrogen Bonds Occupancy

Dynamic cross-correlation map (DCCM) is one of the most effective methods for understanding two-dimensional dynamic information of all the Cα atom fluctuations of proteins [ref]. The effect of either acidic pH or inhibitor-binding, and both effects on the correlated motions of CsCatD2_free_ were investigated (Figure 6). Overall, highly positive correlation regions were observed along the N- and C-lobes and clamp regions. It could be clearly seen that compared with the CsCatD2_free_ at pH 7 (Figure 6A), the correlated motions were substantially increased in the CsCatD2_bound_ at pH 4 (Figure 6D) but slightly increased in the CsCatD2_free_ at pH 4 (Figure 6B) and CsCatD2_bound_ at pH 7 (Figure 6C). Interestingly, acidic pH alone can contribute on the dynamic correlation and, thus, correlated motions of CsCatD2_free_ at pH 4 had similar ones of CsCatD2_bound_ at pH 7. The results were consistent with the RMSD and Rg results. Our finding indicates that both acidic pH and inhibitor-binding could be prerequisite factors for its enzymatic reaction. For the catalytic reaction, the active site pocket needs to be closed which means that movements of the N-lobe and C-lobe should be oriented toward each other, around the pocket.

To explore how CsCatD2_bound_ interacts with the inhibitor under different pH conditions, hydrogen bond and hydrophobic interactions were investigated because hydrogen bonds between inhibitor and surrounding residues are crucial for stabilizing the complex. Noncovalent interactions changed with time during the 50-ns simulation. In particular, acidic pH significantly contributed high hydrogen bond occupancy of the whole simulation time at the Asp_33_ of 94 to 99% and Asp_219_ of 0 to 91% (Figure 7A). CsCatD2_bound_ at pH 4 appeared to adopt a more closed conformation than that at pH 7 due to higher hydrogen bond occupancy at the Asp_219_, Met_297_ and Ile_306_ (Figure 7B).

### 2.8. Twisting of the Active Site Pocket by the Bound Inhibitor and Acidic pH

Flap dynamics imply the time-dependent changes in the physical interactions and conformational flexibilities of the flap region and loops. Based on the structural dynamics, it indicates time-dependent conformational changes in proteins [24,25]. Specific structural metrics have been previously suggested for the flap dynamics of CatD [25]. The distance *d*_1_ (Gly_79_–Met_297_) between the flap tip and hinge residue, TriCα angles, *θ*_1_ (Gly_79_–Asp_33_-Met_297_) and *θ*_2_ (Gly_79_–Asp_219_–Met_297_), which account for the opening and closing of the active site, and dihedral angle *ϕ* (Gly_79_–Asp_33_–Asp_219_–Met_297_) are responsible for the twisting motion (Figure 8A).

The dynamics of the flap structure of CsCatD2 were analyzed. Based on the average structures obtained through a 50-ns MD simulation, each parameter was measured (Figure 8B). The conformational states of CsCatD2 were determined according to the distance *d*_1_ as suggested by Kumalo and Soliman [39]. It was found that the open state was higher than 13 Å, the closed state was lower than 9 Å, and either the semi-open or semi-closed state existed between the two values. In addition to ligand-binding, acidic pH decreased the distance *d*_1_ as follows: from open (13.8 Å) to semi-open (12.6 Å) state in CsCatD2_free_; and from semi-closed (9.7 Å) to closed (8.7 Å) state in CsCatD2_bound_. Compared to the initial structures, the acidic pH and bound inhibitor steered the dihedral angle *ϕ* and reduced the TriCα angles (*θ*_1_ and *θ*_2_) simultaneously. Both the distance *d*_1_ and TriCα angles are likely to decrease. For instance, the dihedral angle *ϕ* of CsCatD2_free_ at pH 7 shifted from +2.1° to −4.7° at acidic pH, to −8.0° for the bound ligand, and to −3.8° for both factors. These results collectively provide evidence of the “twisting” of the binding pocket, which can be seen by the asymmetrical movement of the pocket by the bound inhibitor and acidic pH.

### 2.9. Effect of Acidic pH on Flap Dynamics and Its Correlation with the Twisting Motion of the Active Site Pocket

The time-series distance *d*_1_ was calculated based on the snapshots obtained at 0, 10, 20, 30, 40, and 50 ns during the 50-ns MD trajectories of all systems (Figure 9). The mean and standard error of the distance *d*_1_ were calculated for CsCatD2_free_ at pH 7 (14.85 ± 0.54), CsCatD2_free_ at pH 4 (12.66 ± 0.10), CsCatD2_bound_ at pH 7 (10.52 ± 0.46), and CsCatD2_bound_ at pH 4 (8.69 ± 0.04) (Appendix A). At the beginning of the simulation, the distance *d*_1_ showed semi-open conformation (11.30 Å at pH 7 and 11.29 Å at pH 4) in CsCatD2_free_, whereas *d*_1_ indicated a semi-closed conformation (9.20 Å at pH 7 and 9.66 Å at pH 4) in CsCatD2_bound_. Toward the end of the 50-ns MD simulation, CsCatD2_free_ showed only two open states at pH 4 and then transitioned to a semi-closed state (10.60 Å) at 50 ns. Although the bound forms showed a similar trend at both pH conditions, CsCatD2_bound_ had a lower minimum *d*_1_ of 7.74 Å at pH 4 than the *d*_1_ (8.74 Å) at pH 7 at 50 ns, revealing three semi-closed conformations or three closed states. Notably, the acidic pH induced the distance *d*_1_ of CsCatD2_free_ close to the level of CsCatD2_bound_ at pH 7 around the simulation times (5, 21, 34, and 46 ns) (Figure 10A, Appendix A).

A similar trend was also observed at the dihedral angle *ϕ* in CsCatD2_free_ at pH 4 (Figure 10B). CsCatD2_free_ at pH 7 (average *ϕ*: 2.29 ± 0.43) showed a few small negative values of *ϕ* on an intermittent basis, whereas CsCatD2_free_ at pH 4 (average *ϕ*: −4.68 ± 0.43) made the preferable twisted conformation as represented by a majority of negative values of *ϕ*. Acidic pH induced opposite patterns of *ϕ* in CsCatD2_free_ during 40–50 ns via conformational shifts to the semi-open state (ranging from 16.24 to 5.59) at pH 7 and semi-closed state (from −18.36 to –6.54) at pH 4. The bound forms had negative values of dihedral angles *ϕ*, −7.56 ± 0.38 at pH 7 and −3.78 ± 0.19 at pH 4, throughout the entire simulation time. Among the TriCα angles *θ*_1_ and *θ*_2_ (Figure 10C,D), the angle *θ*_1_ was more consistent with the distance *d*_1_, suggesting that these factors may work together to ensure the timely conformational changes of CsCatD2. Therefore, the three parameters (*d*_1_, *ϕ*, and *θ*_1_) of flap dynamics were correlated with each other to account for the discrepancies in the open or closed states of the binding cleft.

## 3. Materials and Methods

### 3.1. Homology Modeling and Verification

The complete mRNA sequence of CsCatD2 was retrieved from GenBank (Acc. No.: GU433605) [23] and the amino acid sequence was deduced using the Lasergene software (DNASTAR, Madison, Wisconsin, USA). The 3D homology models of CsCatD2_free_ and CsCatD2_bound_ were constructed using Yet Another Scientific Artificial Reality Application (YASARA) Structure v21.6.17 [40] as described by Kang et al. [23]. A hybrid homology model was chosen by combining the best scoring parts of the reliable models for CsCatD2_free_ and CsCatD2_bound_ based on the experimental X-ray-characterized structures of free cathepsin D (PDB code: 5n7n) and inhibitor-bound cathepsin D (PDB code: 5n7q) from the tick *I. ricinus* [28]. The qualities of all models were further evaluated using the Ramachandran plot [41], ERRAT [42], and protein structure analysis (ProSA) [43]. The secondary structure elements were predicted using DSSP (https://swift.cmbi.umcn.nl/gv/dssp/; accessed on 10 August 2021).

### 3.2. Structural Comparison

Structural superpositions were carried out using TM-align [44] with default parameters and the “cealign” method script (https://pymolwiki.org/index.php/Cealign; accessed on 10 August 2021). The TM-score, calculated by TM-align, indicates the fold similarity between the two structures. All structure graphics were prepared using PyMOL (PyMOL Molecular Graphics System, v2.4.1, Schrödinger, LLC, New York, NY, USA).

### 3.3. MD Simulation

All MD simulations were performed using a pre-installed “md_runfast.mcr” macro file within the YASARA Structure v21.6.17 [45] for both the free and bound forms of CsCatD2. All simulations were carried out on a custom-built workstation running Linux Mint 20.1 Ulyssa with Intel Core i7-10700F CPU × 8 (2.9 GHz), random access memory (RAM) of 32 Gb and a graphic processor unit (GPU) of Nvidia GeForce RTX 3090 (24 Gb). The AMBER14 force field was applied under periodic boundary conditions. The simulation cell was allowed to include 20 Å surrounding the protein and filled with water, as a solvent, at a density of 0.997 g/mL. The initial energy minimization was carried out under relaxed constraints using steepest descent minimization in vacuo. To mimic the physiological conditions as previously described [29,30,46], all simulated systems were maintained at pH 4 or pH 7 by adding counter ions to replace the water containing 0.9% sodium chloride (NaCl). Simulations were performed in water at a temperature of 298 K at constant pressure. The cut-off radius for long-range electrostatics was set to 8 Å. The snapshots were saved at 250 picoseconds (ps) intervals for the duration of the 50-ns simulation, as previously suggested [31,32,47].

### 3.4. Post-MD Analysis

Subsequently, snapshots were analyzed using the customized “md_analyze.mcr” and the built-in “md_analyzeres.mcr” macro files within the YASARA package for the RMSD, RMSF, Rg, SASA, and flap parameters, such as the distance *d*_1_ (Gly_79_–Met_297_) between the flap tip and hinge residue, TriCα angles *θ*_1_ (Gly_79_–Asp_33_–Met_297_) and *θ*_2_ (Gly_79_–Asp_219_–Met_297_) for the conformational changes of the active site, and the dihedral angle *ϕ* (Gly_79_–Asp_33_–Asp_219_–Met_297_) for the twisting motion of the binding pocket. The sim format was converted into xtc format using “md_convert.mcr” macro file and, subsequently, into dcd format using Wordom v0.22-rc3 [48]. DCCM and PCA were conducted using the Bio3d package v2.4-2 [49] of R Studio v1.4.1717. Plots were generated using DataGraph v4.7.1 (Visual Data Tools Inc., Chapel Hill, NC, USA) and R Studio.

## 4. Conclusions

Parasitic CatD homologues have gained significant attention for being attractive targets for therapeutic drug design as CatD plays a critical role in the degradation of blood to facilitate the survival of the parasite. Here, we report the pH-dependent structure and flap dynamics of CsCatD2 from the blood-feeding parasite *C. sinensis*, which can cause food-borne parasitic infection and cholangiocarcinoma. The structure–inhibition relationships and validated computational simulations determined in this study will provide important insights into the binding mechanisms of parasitic CatD to potential inhibitors during the pH shift. Especially, acidification of 30–36-ns simulation showed significant contribution to establishment of rigid β-sheets and higher occupancy of hydrogen bonds near the catalytic residue Asp_219_, thereby enabling increased stabilization of the CsCatD2-inhibtor complex. Further, in vitro and in vivo experiments are strongly needed to confirm the structure–function and inhibition.

## Figures and Tables

**Figure 1 pathogens-10-01128-f001:**
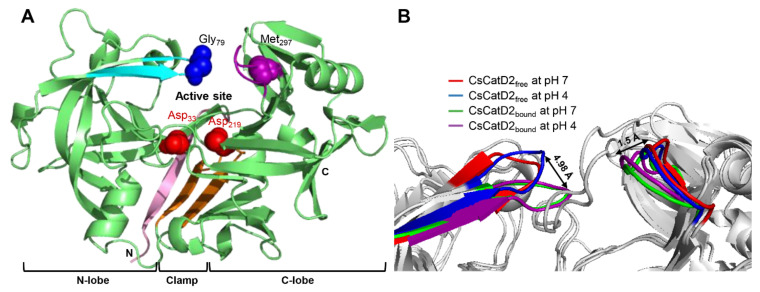
Conserved flap parameters and variations of mature CsCatD2. (**A**) Y flap region, cyan; polyproline loop, purple; two catalytic aspartic residues, red; flap tip residue, blue; hinge residue of the polyproline loop, purple; N-terminal part, light pink; clamp, orange. (**B**) Distances of tip-to-tip and hinge-to-hinge between all CsCatD2 models. Flap tip and hinge residues indicate Gly_79_ and Met_297_, respectively.

**Figure 2 pathogens-10-01128-f002:**
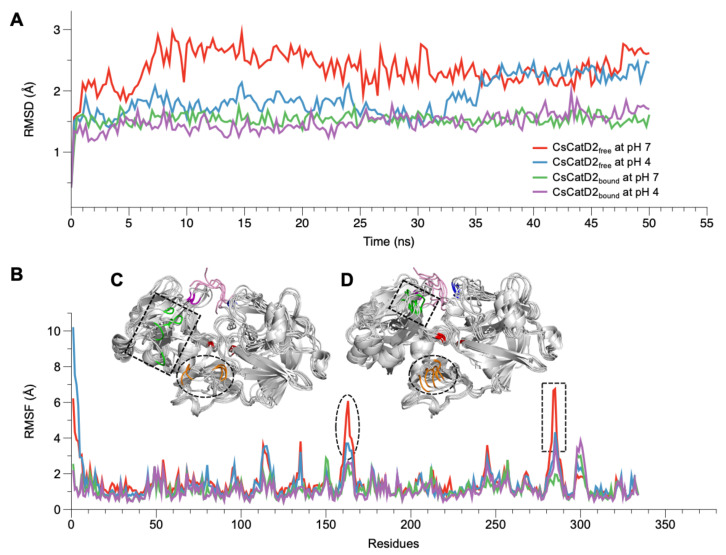
RMSD and RMSF variations of CsCatD2_free_ and CsCatD2_bound_ models at both pH conditions. During the 50-ns MD simulation at neutral (pH 7) and acidic (pH 4) condition, fluctuations of RMSD (**A**) and RMSF (**B**) are shown based on the time and residues, respectively. (**C**,**D**) Structural variations of CsCatD2_free_ at pH 7 (**C**) and pH 4 (**D**) values during the MD simulation. Backbone snapshots of both proteins are in shades of gray, each of which were obtained at 0, 10, 20, 30, 40 and 50 ns. The back view is rotated by 180° about a vertical axis, relative to the view shown in Figure 4A. The dotted circle and rectangle correspond with each region having high RMSF value, such as aa 162–163 (orange part) and aa 284–285 (green part). The remaining color legend is provided in the legend to Figure 4A.

**Figure 3 pathogens-10-01128-f003:**
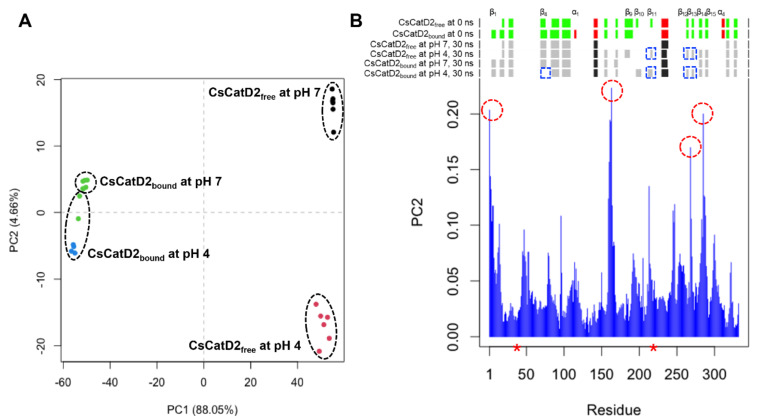
Principal component analysis (PCA) analysis of 28 snapshots during 30–36 ns MD simulation. (**A**) Projection of the motion of CsCatD2_free_ and CsCatD2_bound_ at neutral (pH 7) and acidic (pH 4) condition in phase space along the eigenvectors 1 (PC1) and 2 (PC2). Color labels were assigned to each snapshot, depending on its assigned cluster. (**B**) Contribution of each residue of CsCatD2 to the PC2. Residues showing high contribution are marked by a red circle. Secondary structure elements were predicted using DSSP (https://swift.cmbi.umcn.nl/gv/dssp/; accessed on 10 August 2021) and pH-dependent alterations are marked by a blue rectangle. Colors and alphabets identify α-helices (red and black) and β-strands (green and gray). Asterisk symbols indicate the two catalytic aspartic residues, Asp_33_ and Asp_219_.

**Figure 4 pathogens-10-01128-f004:**
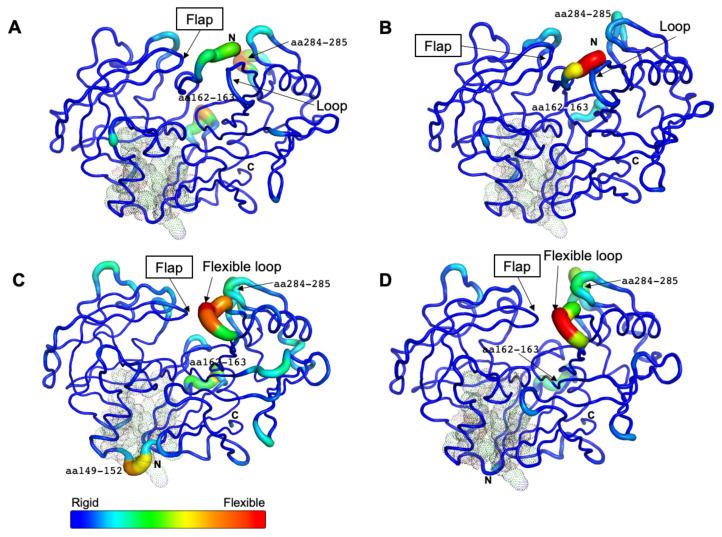
Residual flexibility of the free and bound forms at different pH conditions. The average B-factor-derived mobilities of CsCatD2_free_ at pH 7 (**A**) and at pH 4 (**B**), CsCatD2_bound_ at pH 7 (**C**) and pH 4 (**D**) were visualized. All residues (at positions 14–16, 32, 97, 149, 170, 171, and 173) of potential allosteric inhibitory sites are indicated with black dots (see Section 2.5).

**Figure 5 pathogens-10-01128-f005:**
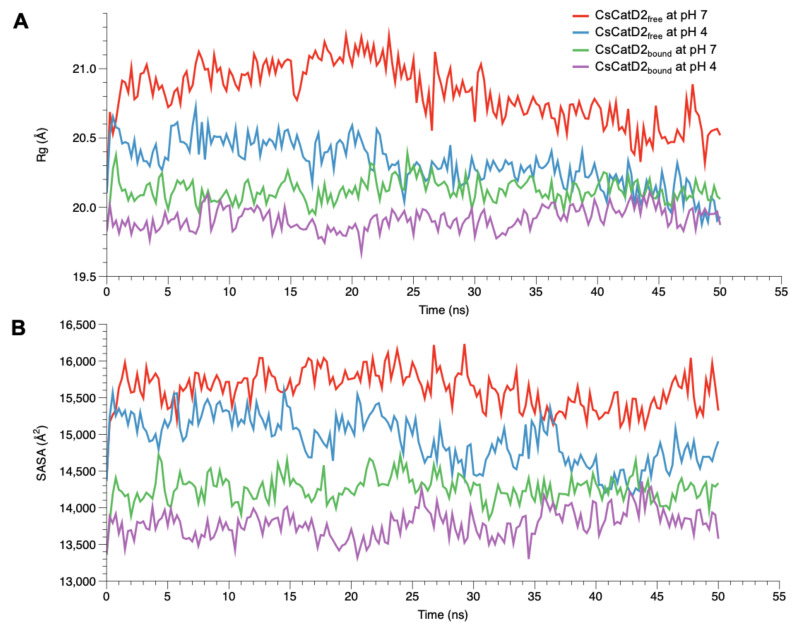
Conformational changes of free and bound CsCatD2 at both pH conditions. Rg (**A**) and SASA (**B**) of CsCatD2_free_ and CsCatD2_bound_ were calculated at neutral (pH 7) and acidic (pH 4) condition throughout the trajectory of 50 ns.

**Figure 6 pathogens-10-01128-f006:**
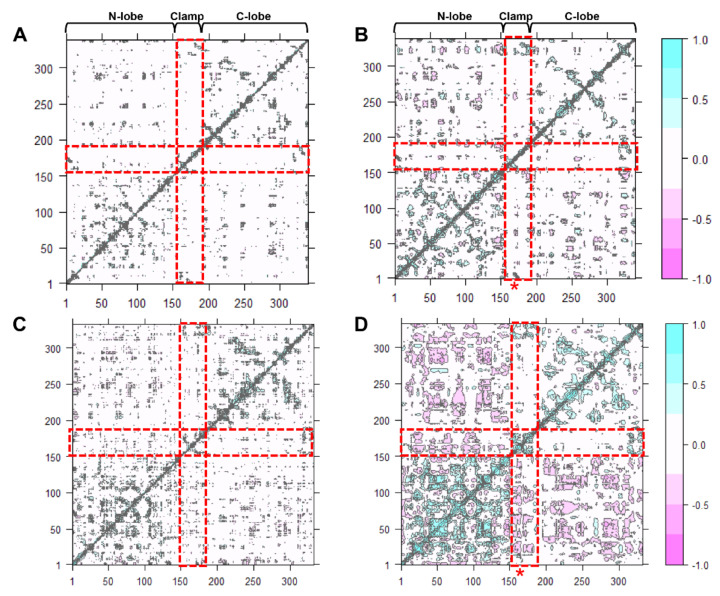
Dynamic cross-correlation map of the Cα atoms of CsCatD2 during 50-ns MD simulation. CsCatD2_free_ at pH 7 (**A**) and at pH 4 (**B**), CsCatD2_bound_ at pH 7 (**C**) and pH 4 (**D**). The cyan color indicates positive correlated motions, and the pink color shows negative correlated motions. Red rectangles depict N-lobe (aa 1–152), clamp (aa 153–172) and C-lobe (aa 173–332). Asterisk symbols indicate the two catalytic aspartic residues, Asp_33_ and Asp_219_.

**Figure 7 pathogens-10-01128-f007:**
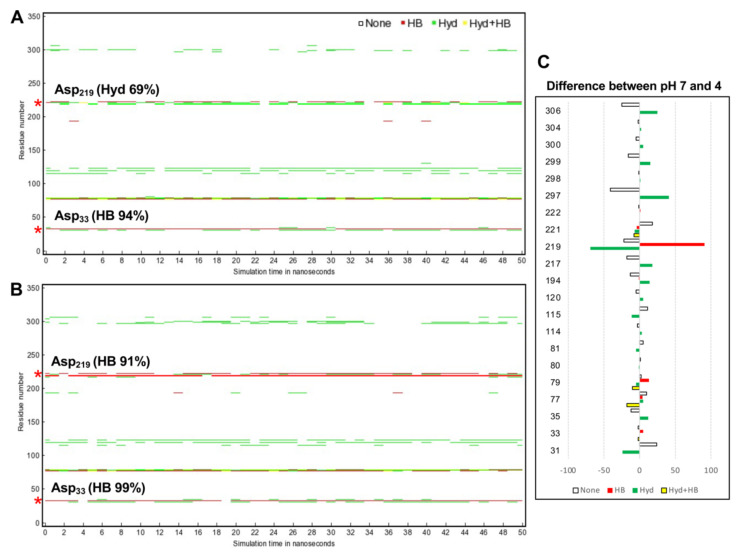
Protein–inhibitor contact plot with per-residue interactions of CsCatD2_bound_ at pH 7 (**A**) and pH 4 (**B**) during 50-ns MD simulation. (**C**) Residue list shows fluctuations of per-residue contacts by subtraction of the value for CsCatD2_bound_ at pH 4 (right) from the corresponding value at pH 7 (left). HB, hydrogen bonds in red; Hyd, hydrophobic contacts in green; Hyd+HB, both hydrogen bonds and hydrophobic contacts in yellow. Asterisk symbols indicate the two catalytic aspartic residues, Asp_33_ and Asp_219_.

**Figure 8 pathogens-10-01128-f008:**
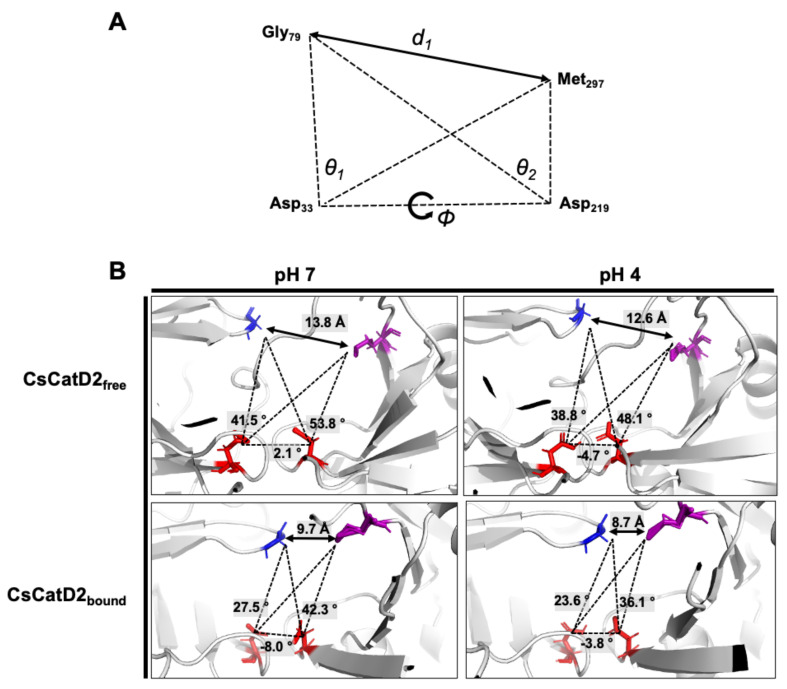
Schematic representation to define the flap–structure dynamics in the free and bound CsCatD2. (**A**) Parameters used to describe the flap dynamics of the active site and flap region: the distance (*d*_1_) between the flap tip and flexible region hinge residue, the dihedral angle (*ϕ*) and the TriCα angles (*θ*_1_ and *θ*_2_). (**B**) Parameter values obtained from the average free and bound CsCatD2 during MD simulations at neutral (pH 7) and acidic (pH 4) conditions. Four residues in active sites are shown in sticks.

**Figure 9 pathogens-10-01128-f009:**
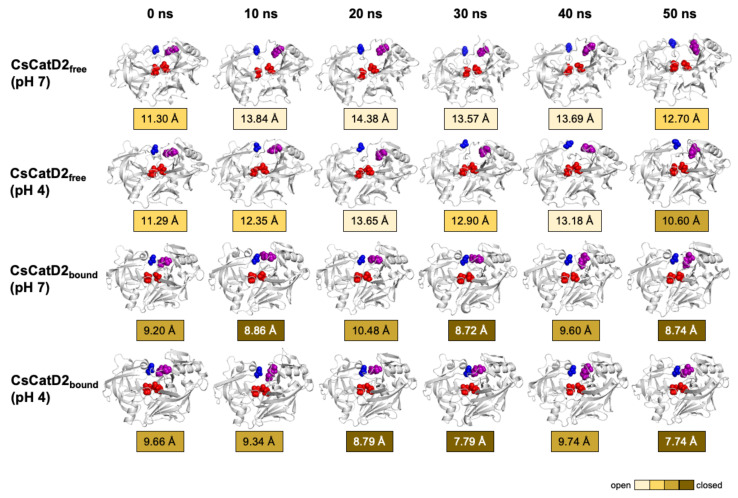
Time-series snapshots of *d*_1_, the distance between the flap tip and hinge residue, throughout the 50-ns MD simulation. Structural changes of the distance *d*_1_ of CsCatD2_free_ and CsCatD2_bound_ were visualized based on the six time points 0, 10, 20, 30, 40 and 50 ns. The conformational states were determined according to the distance *d*_1_ as suggested by Kumalo and Soliman [39], such as open state of *d*_1_ > 13 Å, closed state of *d*_1_ < 9 Å and semi-conformational state of 9 ≤ *d*_1_ ≤ 13. Four residues in active sites are shown by spheres in the same color as in Figure 1A.

**Figure 10 pathogens-10-01128-f010:**
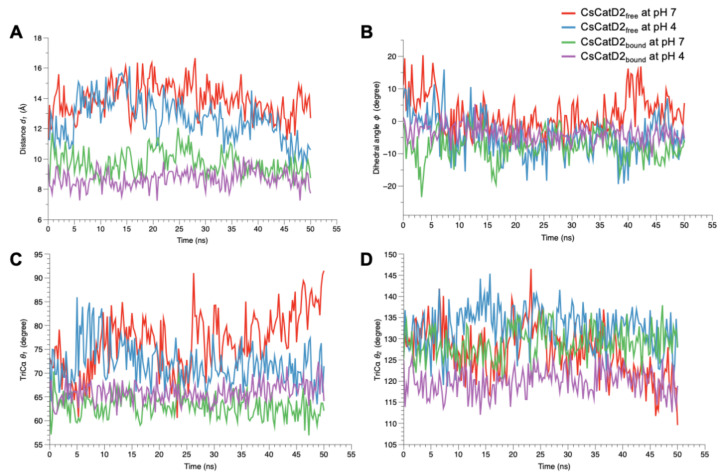
Flap dynamics of free and bound CsCatD2 throughout the trajectory of 50 ns. Flap parameters of CsCatD2_free_ and CsCatD2_bound_ were determined according to both pH conditions, pH 7 and pH 4. (**A**) distance *d*_1_, (**B**) dihedral angle *ϕ*, (**C**) TriCα angle *θ*_1_ and (**D**) TriCα angle *θ*_2_.

## Data Availability

All data generated or analyzed during this study are included in this published article.

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
