# Peer review of "pH-Dependent Structural Dynamics of Cathepsin D-Family Aspartic Peptidase of Clonorchis sinensis"

_pathogens, 2021, doi:10.3390/pathogens10091128_

Round 1
Reviewer 1 Report
This study is a continuation of the research begun by the authors earlier and is devoted to characterization of cathepsin D family aspartic peptidase of the carcinogenic liver fluke Clonorchis sinensis. The authors present the data using the computational analysis of the pH-dependent structural dynamics of cathepsin D isozyme 2. Results of the study potentially might be useful for design of an inhibitor of the enzyme.
The main drawbacks of the manuscript are (i) the lack of a convincing justification why CsCatD2 is a promising drug target for therapy of C. sinensis infection; (ii) too dry and formalized presentation of the results and (iii) the lack of full discussion of the data obtained.
Introduction and conclusion of the manuscript are very speculative. Scientific novelty of the research is presented extremely vague. It is not clear how unique the characteristics obtained by the authors are. In addition, it is unclear whether it is possible, based on these data, to suggest a selective inhibitor of the enzyme.
I recommend that the authors completely rewrite the manuscript and pay particular attention to the Discussion of Results section.
Reviewer 2 Report
In this manuscript, Kang et al., report analysis of the molecular dynamics simulations performed on Clonorchis sinensis cathepsin D isozyme 2 (CsCatd2). They examine the ligand (pepstatin, inhibitor) free and bound forms of the enzyme under acidic (pH 4) and neutral (pH 7) conditions to understand the conformational transitions associated with the biochemical activity and potential allosteric regulation.
- The manuscript is well written and easy to follow but the figure quality in the pdf available for peer-review is extremely poor. I believe the results reported in the manuscript are well aligned with the conclusions but I would like to see better-quality figures.
- Section 2.1: “The IrCD1 structures were chosen as the best templates via the protein-Basic Local Alignment Search Tool (BLASTP) against the PDB database”. Provide details of the sequence similarity and any other parameters to indicate the quality of the match between the CsCatD2 and IrCD1.
- Section 2.2: “The intermolecular interactions between CsCatD2….flap residue Gln77”. Add a supplementary figure to show the comparison of intermolecular interactions of CsCatD2 and HsCatD with inhibitors.
- Section 2.4: “At these specific points-of-time, the RMSD was abruptly increased by 0.8 Å during which a transition in the tertiary structure may have occurred”. It will be interesting to retrieve the snapshots around this time-point and describe the tertiary structural changes in detail.
- Section 2.6: “This highly mobile region was located at the boundary of the potential allosteric inhibitory pocket” and later “neutral pH can induce the opening of the allosteric inhibitory site of CsCatD2 similar to HsCatD and IrCatD”. Can you clarify if the allosteric pocket is well established in the HsCatD and IrCatD and add the relevant description in the text?
- Section 2.7: “The low Rg and SASA values throughout the remainder of the simulations were also consistent with the adoption of a new conformation.” It would be helpful to carefully look at the structural snapshots and show structures of major/stable conformations observed during the MD simulations.
- I recommend making the coordinates of the homology models and the major structural snapshots (e.g. the conformational states as in Table S2) available to the public (perhaps as downloadable supplementary files). Availability of the coordinates will increase the reach and impact of the results presented in this manuscript and the trend will likely increase the engagement of researchers from the biochemistry, structural biology, and computational biology fields.
Round 2
Reviewer 1 Report
The revised version of the manuscript contains only a few modifications. In fact, the authors politely commented on my questions, but did not answer them. I have no doubt that the results obtained by the authors have potential value.
Aspartic cathepsin D-like peptidases of many endo- and ectoparasites have significant potential as molecular targets for antiparasitic therapy. It has been shown experimentally that suppression of the activity of these enzymes leads to: growth arrest, morphological alterations (Trypanosoma cruzi); reduced parasite viability (Plasmodium falciparum); reduced survival of blood flukes (Schistosoma japonicum) etc. In the list of parasites for the survival of which cathepsin D-like peptidases are important, there are still no liver flukes belonging to the Opisthorchidae family. Ov-APR-1, an aspartic protease from the carcinogenic liver fluke, Opisthorchis viverrini was characterized in 2009; partial characterization of two cathepsin D family aspartic peptidases of Clonorchis sinensis was performed in 2019. In both studies, the effect of inhibiting the enzyme activity on the vital activity of helminths was not studied. In this regard, I had a question for the authors – why do you think that this enzyme is a good therapeutic target for the treatment of clonorchiasis?
I also leave all the questions that I had earlier and once again recommend the authors should revise the manuscript. I suggest omitting all speculations regarding the significance of C. sinensis cathepsin D as a drug target for therapy of clonorchiasis. I also suggest the authors hypothesize about functioning of the enzyme and its role in numerous parasitic mechanisms, from protein trafficking to tissue penetration, immune evasion, and digestion of host blood proteins.
Reviewer 2 Report
For query 5 (section 2.6): It is still not clear whether the allosteric pocket is a "potential allosteric inhibitory pocket" or has it been experimentally shown that this pocket can bind ligands? I suggest authors expand on the relevant results from references 32 and 38 and any other literature that supports the existence of the allosteric pocket.
Round 3
Reviewer 1 Report
The authors have addressed the comments and suggestions I made in the first review. The quality of the manuscript has significantly improved.
Author Response
Thanks to your insightful and constructive comments, our manuscript could be significantly improved. We are sincerely grateful to you for your invaluable help.